# An Emergent Form of Cardiotoxicity: Acute Myocarditis Induced by Immune Checkpoint Inhibitors

**DOI:** 10.3390/biom11060785

**Published:** 2021-05-22

**Authors:** Roberta Esposito, Teresa Fedele, Silvia Orefice, Vittoria Cuomo, Maria Prastaro, Mario Enrico Canonico, Federica Ilardi, Francesco De Stefano, Ludovica Fiorillo, Ciro Santoro, Giovanni Esposito

**Affiliations:** 1Department of Clinical Medicine and Surgery, Federico II University Hospital, 80131 Naples, Italy; teresafedele88@libero.it (T.F.); silviaorefice88@gmail.com (S.O.); vittycuomo@yahoo.it (V.C.); ludovica.fiorillo91@gmail.com (L.F.); 2Mediterranea Cardiocentro, 80122 Naples, Italy; fedeilardi@gmail.com; 3Department of Advanced Biomedical Sciences, Federico II University Hospital, 80131 Naples, Italy; prastaro@unina.it (M.P.); mecanonico@me.com (M.E.C.); ciro.santoro@unina.it (C.S.); espogiov@unina.it (G.E.); 4Division of Cardiology, Villa dei Fiori, 80011 Acerra, Italy; fradestefano@alice.it

**Keywords:** myocarditis, cancer, immune checkpoint inhibitors, immune-related adverse events, cardio-oncology

## Abstract

Immune checkpoint inhibitors (ICIs) are monoclonal antibodies that activate the immune system, aiming at enhancing antitumor immunity. ICIs have shown great promise in the treatment of several advanced malignancies. However, therapy with these immunomodulatory antibodies may lead to a wide spectrum of immune-related adverse events in any organ and any tissue. Cardiologic immune-related events include pericarditis, pericardial effusion, various types of arrhythmias including the occurrence of complete atrioventricular block, myocardial infarction, heart failure, and myocarditis. Although relatively rare, myocarditis is associated with a very high reported mortality in comparison to other adverse events. Myocarditis often presents significant diagnostic complexity and may be under-recognized. When confronted with an unexpected change in the clinical picture, the physician must differentiate between immune-related adverse events, cancer worsening, or other causes unrelated to the cancer or its therapy. However, this is not always easy. Therefore, with the increasing use of checkpoint inhibitors in cancer, all providers who care for patients with cancer should be made aware of this rare, but potentially fatal, cardiologic immune-related adverse event, and able to recognize when prompt consultation with a cardiologist specialist is indicated. In this review, we evaluate currently available scientific evidence and discuss clinical manifestations and new potential approaches to the diagnosis and therapy of acute myocarditis induced by ICIs. Temporary or permanent discontinuation of the ICIs and high-dose steroids have been administered to treat myocarditis, but symptoms may worsen in some patients despite therapy.

## 1. Introduction

Cancer therapies have continuously evolved over the recent years. Oncologic researches have aimed at finding the most effective anti-tumoral drugs, and the result of this effort is a dramatic change in the natural history of the vast majority of neoplasms nowadays, in respect of the past decades. It has been widely demonstrated that almost all chemotherapic drugs currently used in clinical practice can determine cardiovascular toxicity, such as heart failure, coronary artery disease, and arrhythmias. Considering the impact of effective therapies on the natural history of the cancer diseases, the years of survival for cancer patients are growing [1], and so the patient can develop forms of cardiotoxicity that were not observed in recent decades due to more limited survival [2]. This evidence, indeed, raises the need to develop strategies and techniques (clinical, laboratory, and echocardiographic) to detect the initial forms of cardiotoxicity [3]. Immune checkpoint inhibitors (ICIs) have represented the paradigm of oncologic treatment in the last decade. Among the many immunotherapeutic strategies, immunotherapy with checkpoint inhibitors has proven as an effective treatment for non-small cell lung cancer and melanoma. However, the indications for treatment with PD-1/PD-L1 checkpoint inhibitors are rapidly increasing, as clinical trials continue to show their efficacy in more wide range of solid tumors. Targets immune checkpoints are programmed death-1 (PD-1), its ligand, programmed death ligand-1 (PD-L1), and cytotoxic T lymphocyte antigen-4 (CTLA-4). There are currently seven Food and Drug Administration-approved ICIs: anti–PD-1 antibodies (Nivolumab, Pembrolizumab, Cemiplimab), anti–PD-L1 (Atezolizumab, Avelumab, Durvalumab), or anti–CTLA-4 antibodies (Ipilimumab). Hundreds of clinical trials are at present underway around the world to evaluate the efficacy and safety of ICIs, as monotherapy or in combination [4].

In normal conditions, T-cells can recognize the antigens expressed on the surface of the self-cells through different pathways, such as Programmed Death protein—1 (PD-1) on the T-cell which recognizes its ligand (PDL1) on the host cell. The PD1–PDL interaction has negative control and helps the activated T-cell from attacking the normal host cells. Therefore, the expression of PD-L1 in the tumor micro-environment protects cancer cells from immune-mediated destruction. Another similar mechanism involves the Cytotoxic Lymphocytes Antigen-4 (CTLA-4) on T-cell, which binds to B7 protein expressed on the surface of APC. This interaction blocks the T-cell activation [5].

Therefore, ICIs are monoclonal antibodies that can break up the immunologic homeostasis and reduce T-cell tolerance. However, a substantial number of patients do not benefit from ICIs, and some patients develop even severe side effects. Nevertheless, ICIs seem to have a better safety profile than chemotherapy [6,7]. Finding predictive biomarkers for differentiating responders and non-responders would allow not only to improve treatment outcomes, but also to reduce adverse effects. Patient-derived xenograft models have the potential to predict individual responses to drugs and treatments. However, further studies are needed to develop strategies to investigate the effects of ICIs, because patient-derived xenografts models can only be established in immunocompromised mouse strains [8]. This is an important topic. In fact, despite their great therapeutic results, ICIs are not free from undesired effects and toxicity. So, this promising form of therapy can cause a new form of toxicity: the immune-related adverse events (IRAEs) [9], that depend on the unwanted immune aggression of the host cells.

Clinical and observational studies have demonstrated several IRAEs in patients treated with ICIs, i.e., colitis, pneumonitis, cutaneous reactions, hepatitis, endocrinopathies, nephritis, encephalitis, and others. The incidence of fatal IRAEs has ranged from 0.3% to 1.3%, and the IRAEs most commonly associated with fatality include myocarditis, pneumonitis, encephalitis, and hepatitis [10]. ICI-related myocarditis are the most dangerous IRAEs, whose incidence has been reported between 0.09% and 2.4% [11,12], with a fatality rate estimated at 36–60% [13].

Besides myocarditis, immune-mediated therapy can cause various forms of cardiotoxicity [14], including pericarditis, pericardial effusion, cardiac arrhythmia, myocardial infarction, heart failure, Takotsubo cardiomyopathy, and cardiac arrest. It is emphasized that the incidence of cardiovascular complications depends on which drug and therapeutic protocol (monotherapy and/or combination) are used [11,12,15,16] (Table 1 and Table 2).

The tables below show the incidence of cardiotoxic events reported to be associated with the use of ICIs [22], with particular attention to myocarditis. Most of the data available in the literature are derived from case reports, case series, and retrospective studies. As result, these numbers reported do not reflect the true incidence of these complications. In addition to this, only the most severe forms may have been reported, leaving out the subacute, paucisymptomatic, undiagnosed, or self-limited forms. Thus, the incidence of myocarditis could be higher. Interestingly, an observational, retrospective, pharmacovigilance study describes reports of cardiovascular IRAEs in patients who received ICIs from 2008 to 2018, through analysis of VigiBase, the World Health Organization database for autoimmune adverse drug reactions (ADRs) [23].

This study compares the proportion of selected adverse drug reactions reported for a single drug or group of drugs (e.g., ICIs) with the proportion of the same adverse drug reactions for a control group of drugs (e.g., full database), through a case–non-case analysis and using the reporting odds ratio (ROR) and the information component (IC) [18]. The results showed that ICI can cause cardiovascular toxicities that include myocarditis, pericardial disease, vasculitis, including temporal-arteritis with a risk for blindess and arrhythmias. They identified a significant incidence of ICI-associated myocarditis characterized by early onset and high fatality rate (50% of cases). ICI-associated myocarditis was associated with different ICI and many cancers. The only cancer-related risk factor was the combination ICI treatment.

Histological examination of myocardial tissue in patients with ICI-associated myocarditis demonstrates an infiltration consisting mostly of lymphocytes and macrophages, within both the myocardium and conduction system (including atrioventricular nodes) [11].

PD-1/PD-L1 signaling plays a cardio protective role against uncontrolled immune response. Indeed, PD-L1 expression in the heart appears to be tightly regulated. In support of this, a study demonstrates that mice with a genetic loss of PD-L1 can develop spontaneous myocarditis with dense infiltrates of T cells [24,25].

## 2. Screening and Surveillance

Guidelines or screening plans do not exist for cardiotoxicity monitoring of ICI-associated myocarditis. Guidelines by the American Society of Clinical Oncology regarding the management of IRAEs recommend a baseline electrocardiogram (ECG) and assessment of a baseline troponin, but there is no stringent protocol for surveillance during anti-cancer therapy.

We would propose a cardiotoxicity monitoring strategy which includes, in the beginning, a baseline assessment of risk factors for ICIs related myocarditis.

Risk factors for myocarditis ICI-related before anti-cancer therapy include diabetes, sleep apnea, and a higher body mass index [12].

There is currently no evidence demonstrating a vulnerability to the development of ICI-related cardiotoxicity in patients with previous cardiovascular disease, while previous exposure to chemotherapeutics such as anthracyclines may increase the risk of myocarditis [12,26].

Combination therapy detects a high risk for ICIs related myocarditis (it has been reported that the risk is fivefold in nivolumab and ipilimumab compared with nivolumab only) [11].

On the other hand, a recent study regarding influenza vaccination and myocarditis among patients receiving ICIs suggests that influenza vaccination may be protective [18]. Patients that received influenza vaccination had lower rates of ICI-related myocarditis compared to non-vaccinated patients [27].

Furthermore, because cancer patients often present vitamin D deficiency, examination of vitamin D status can be used for risk assessment of IRAEs. So, there is emerging consideration of administering concomitant vitamin D with checkpoint inhibitors to alleviate IRAEs [28,29].

Patients who are at high risk for the development of ICIs related myocarditis need a monitoring strategy to be planned with cardiologists paying close attention to the first week of treatment. Most cases occur early, approximately after the first or second ICI infusion [18].

During ICIs therapy, a cardiotoxicity monitoring plan may include evaluation of troponins (troponin-I and -T) and ECG before each cycle, together with evaluation of left ventricular ejection fraction (LVEF) and global longitudinal strain (GLS) at baseline echocardiography and then at months interval.

Also, the presence of accompanying clinical conditions, such as myositis, colitis, or other IRAEs, can be considered a predictor of myocarditis [11,30].

The presence of other IRAEs can help the clinician to suspect ICI-associated myocarditis.

## 3. Diagnosis

The diagnosis and management of ICI-related myocarditis represents a clinical challenge. The determination that myocarditis is related to ICI therapy should be made by assessment of the temporal link, excluding all the other causes of acute myocardial dysfunction [31]. ICI-related myocarditis can present with various and heterogeneous symptoms, such as dyspnoea, palpitations, chest pain, asthenia, generalized malaise, nausea, weight gain, fever, and cough. The diagnostic algorithm may begin with troponin screening or due to the onset of symptoms during ICI therapy. However, troponins are not specific to myocarditis, but a gradual normalization of troponins level correlates with the clinical response to immunosuppressant therapy [32,33].

Another common laboratory test that could raise suspicion of myocarditis is BNP. BNP is not a specific biomarker for ICI-related myocarditis due to false-positive results in common cancers but it may be useful for monitoring. Therefore, BNP should be determined as a baseline value [34].

Similar to the troponin assay, ECG is part of the diagnostic protocol and surveillance [35]. ECG may register intraventricular conduction delay, PR interval prolongation, complete heart block [11], and other forms of arrhythmia, including atrial fibrillation and ventricular arrhythmias [14].

Echocardiographic exanimation is the first-line examination in the suspicion of ICI-related myocarditis. However, the LVEF could be normal even in fulminant forms and it is important to underline that a normal ejection fraction does not exclude the diagnosis of myocarditis [36].

As a result, a baseline collection of the clinic, ECG, and echocardiography is relevant as the symptoms, signs, and laboratory data of ICI-related myocarditis can be attributed to a broad spectrum of cardiovascular diseases, including arrhythmias, acute and chronic coronary syndrome, and left ventricular dysfunction. ECG, echocardiography, and serial measurement of troponins during ICI therapy are fundamental aspects to detect the dynamics of their changes, facilitating early diagnosis of ICI-related myocarditis versus other cardiac disorders [34,37].

Newer echocardiographic techniques, such as speckle tracking echocardiography-derived GLS can detect the initial reduction of the LV longitudinal contractile function.

Compared to LVEF alone, an increasing number of studies suggest the use of GLS as an early predictor of myocardial damage secondary to cardiotoxicity, particularly in asymptomatic patients with preserved LVEF [36].

In acute myocarditis, the global longitudinal function is often affected [38]. In a study in patients with acute myocarditis, GLS appears to be related to the amount of myocardial edema and the exact regional localization detected by CMR [39].

Therefore, the measurement of GLS should be applied in all suspected or certain cases of myocarditis.

A recent large international multicenter study presents first data about the GLS’s role among patients with ICI-related myocarditis. This study proves that: 1. GLS measured pre-ICI therapy was similar between cases (ICI-induced myocarditis cases) and control (subjects on ICI therapy without myocarditis); 2. GLS decreased in patients who developed myocarditis, but not in those who did not develop it; 3. GLS was reduced in ICI-related myocarditis cases, presenting with either a preserved or reduced ejection fraction; 4. Lower GLS was a predictor of MACE in patients with both preserved and reduced EF [26].

It is specified that the term MACE is intended for major adverse cardiac events, a composite of cardiovascular death, cardiogenic shock, cardiac arrest, and complete heart block [12]. The innovation of the mentioned study is to attribute a prognostic value to GLS where each 1% reduction was associated with a 1.5-fold increase in MACE among cases with reduced EF and a 4.4-fold increase in MACE in those with a preserved EF.

CMR plays a major role in the diagnosis of acute myocarditis. CMR is the gold-standard non-invasive imaging test for diagnosis and risk prediction in myocarditis because it identifies functional and structural myocardial abnormalities and indirectly characterizes underlying histopathological changes in the myocardium [40,41].

In 2018, an expert consensus paper proposed an update of Lake Louise Criteria for magnetic resonance diagnosis of myocarditis. Two major criteria are proposed to diagnose myocardial inflammation: 1. T2-based marker (T2-weighted imaging or T2 mapping) for myocardial edema because edema is an essential component of acute or active inflammation; 2. T1-based marker (LGE, T1 mapping, or ECV) for associated myocardial injury. One of the two major criteria is sufficient to make the diagnosis, but the specificity is increased if CMR scan demonstrate both of them [40]. LGE might also serve to differentiate between acute myocarditis and acute myocardial infarct. In the first case, it is almost always located in the epicardial/mid myocardial layer, sparing the subendocardial region in a nonischemic distribution, while in the second case it is typically seen in the subendocardial area [42]. Although LGE may identify myocardial fibrosis/scar considered as a sequel of myocardial inflammation in myocarditis, fibrosis may develop and accumulate after becoming detectable on CMR or biopsy. Currently, there are scarce data about the use of CMR in ICI-associated myocarditis which correlates with EMB.

For example, the absence of LGE and the absence of increased T2-weighted STIR signal on a CMR do not rule out the potential myocarditis diagnosis because late gadolinium enhancement and T2-weighted STIR imaging are expressions of local fibrosis or inflammation that haven’t become qualitatively apparent yet.

So, increased time between clinical suspicion and CMR is associated with greater detection of LGE. Based on literature searches and data collection of available studies, caution is required if using an LGE or qualitative T2-weighted STIR imaging approach to exclude ICI-associated myocarditis [43].

In the literature, a case of immunotherapy-induced myocarditis is described with LGE on CMR correlating with areas of myocardial lymphocytic infiltrate and fibrosis on post-mortem pathology. It is the first case demonstrating the direct histological correlation of T-lymphocytic infiltration with areas of LGE on CMR. So, we emphasize the need to improve diagnostic techniques that will allow us to make an earlier diagnosis of myocarditis and guide therapy in these patients [44].

Indeed, endomyocardial biopsy has been suggested to be the gold standard for diagnosis of ICI-related myocarditis [35].

According to the Dallas criteria, acute myocarditis diagnosis is defined by lymphocytic infiltrates in association with myocyte necrosis [45].

However, we must consider that it is an invasive procedure with false-negative results (e.g., when the biopsy material is taken from disease-free territories). In one study including 38 patients, using autopsy as the gold standard, the sensitivity and specificity of EMB were about 60% and 80%, respectively [46].

In the absence of specific recommendations for patients with suspected immune-related myocarditis, we believe that the indications to perform EMB are those given in the last consensus [40]. According to the recommendations of the American Heart Association, the American College of Cardiology, and the ESC, the indication for EMB should be considered for patients with acute (<2 weeks), severe new onset heart failure with hemodynamic compromise, as well as new onset heart failure (between two weeks and three months) with a dilated left ventricle and new ventricular arrhythmias, atrioventricular block II to III, or failure to respond to medical therapy and usual care within 1–2 weeks [47]. For patients with an infarction-like presentation, the ESC Working Group statement recommends EMB after the exclusion of coronary heart disease [48], whereas more recent heart failure guidelines recommend CMR to identify myocarditis in patients with suspected or established heart failure [49]. EMB should also be considered in patients with persistently elevated troponin values and progressive cardiac dysfunction despite maximal heart failure therapy. The pre-procedural localization of inflammatory changes in CMR images may reduce sampling errors and improve therapeutic decision making and prognostication [50,51].

Bonaca et al. proposed to categorize suspected cases of myocarditis ICI-related into three groups including definite myocarditis, probable myocarditis, and possible myocarditis, based on clinical, laboratory, and instrumental data [28] (Figure 1).

## 4. Treatment Options

Currently, the treatment of ICI-associated myocarditis consists of immunosuppression and symptomatic heart failure therapy, such as diuretics, beta-blockers, angiotensin-converting enzyme inhibitors, or aldosterone receptor blockers, angiotensin receptor-neprilysin inhibitor, SGLT2 inhibitors depending on the severity of presentation.

The management of cardiotoxicity demands close collaboration between oncologists and cardiologists, especially regarding the decision to discontinue immunotherapy. There is limited evidence regarding the best management of cardiac toxicity because of the lack of comparative studies among discontinuation and continuation of ICI in the case of myocarditis. Anyway, due to the high mortality rate of myocarditis, guidelines give strong recommendations about the priority of suspending therapy in case of myocarditis induced by ICIs. The first-line treatment for ICI-related myocarditis is corticosteroid: there are limited and variables data in terms of initial corticosteroid dose and treatment strategies. In retrospective studies, higher initial dose (i.e., intravenous methylprednisolone 1–2 mg/kg/die or 1000 mg/die in severe myocarditis, to be titrated in the following 4–6 weeks) and earlier initiation of corticosteroids (preferably within 24 h) are associated with improved cardiac outcomes. The duration and titration should be based on clinical improvement and troponins levels, as well as the presence of other IRAEs. Usually, the duration of treatment should last 3–6 months [35]. Myocarditis appears to have the highest risk of death (about 40%) compared to other IRAEs and, sometimes, corticosteroids are not sufficient [10].

Refractory ICI-associated myocarditis remains of life-threatening toxicity, is associated with high mortality, and it thus demands the use of other therapeutic strategies.

In severe refractory cases to steroids, additional strategies could be Plasmapheresis and/or intravenous immunoglobulin IVIG (2 g/day) and additional immunosuppressive agents could include Mycophenolate Mofetil (1 g oral BD), Tacrolimus, Infliximab, and anti-thymocyte globulin (ATG). Other investigational second-line agents are Abatacept, Alemtuzumab, Tofacitinib, and Tocilizumab.

Recommendations regarding these second-line agents are weak, due to poor data on their use in myocarditis based only on case reports and few retrospective studies.

ATG is used to prevent and treat acute rejection in organ transplantation and it has also been used in patients with cardiotoxicity, because of histological similarity between ICI-associated myocarditis and cardiac transplantation rejection. The underlying mechanism could be associated with the rapid reduction in lymphocyte infiltration and T cell activation in cardiac tissue.

A case report regarding a patient with heart failure after combination immunotherapy (ipilimumab at 3 mg/kg and nivolumab at 1 mg/kg) showed stabilization of hemodynamic status and significant improvement of cardiac function (LVEF from 20% to 40%) thanks to ATG, given following the institutional heart transplant rejection protocol [52].

Another study shows that two patients, with myocarditis and clinical worsening during steroid treatment, responded well to ATG with rapid remission of cardiogenic shock and malignant arrhythmias [53].

However, more data are needed to demonstrate the safety and the efficacy of this approach.

Recently it has been reported the successful use of Abatacept and Alemtuzumab, two selective immunosuppressive drugs, for the treatment of ICIs related myocarditis. Abatacept is a fusion protein between the extracellular domain of CTLA-4 and the Fc domain of IgG, which binds to CD 80–86 and inhibits CD28-mediated costimulation of T-cell. A study reports a case concerning a patient with steroid-refractory Nivolumab-associated myocarditis and myositis, who responded successfully to intravenous Abatacept (at a dose of 500 mg every two weeks, for a total of five doses) [54].

Alemtuzumab, a monoclonal antibody that binds to CD52, promotes the complement-mediated lysis of peripheral immune cells (monocytes, lymphocytes, macrophages, natural killer cells, and dendritic cells). A recent case report is about a 71-year-old woman, who presented myositis and myasthenia gravis related to Pembrolizumab, which had an initial good response to immunosuppressive treatment with Methylprednisolone and Rituximab. However, she developed life-threatening cardiac arrhythmias 18 days later, which had a rapid resolution with a single dose of 30 mg of Alemtuzumab [55]. In any case, evidence about the use of Alemtuzumab in patients with myocarditis and other iRAEs is still limited.

Infliximab, a chimeric monoclonal antibody against TNF-α, is also used in severe steroid-refractory myocarditis with significant clinical recovery and biochemical normalization [53]. It is important to remember that infliximab, as per the datasheet, is contraindicated for patients with moderate-severe heart failure (Class NYHA III-IV). A randomized phase II study, assessing the safety of infliximab in patients with moderate to severe heart failure, has shown a clinical worsening in patients treated with infliximab at high doses (10 mg/kg) compared to placebo [56].

Tocilizumab, an anti-IL-6R agent, may be used for rapid resolution in refractory high-grade myocarditis. A study reports a case of corticosteroid refractory myositis and myocarditis (arrhythmias, third-grade atrioventricular block, and elevated concentrations of troponins I and T) in a 57-year-old male, who was in treatment with combination therapy (Nivolumab and Ipilimumab). He was treated with intravenous Tocilizumab at a dose of 8 mg/kg body weight, weekly, for a total of two doses, and symptoms of myocarditis (arrhythmias) and myositis (muscular weakness and pain) progressively disappeared [57].

Moreover, IL-6 signaling promotes a pro-tumorigenic immune-suppressive network. Studies show that tumor cells produce IL-6, which inhibits the maturation of DC and reduces Th1 differentiation of CD4+ T, resulting in impaired adaptive immune responses against the tumors. IL-6 also stimulates the production of immune-suppressive factors such as IL10, and increases the production of vascular endothelial growth factor (VEGF) by myeloid cells, promoting tumor vascularization [57]. Therefore, Tocilizumab, thanks to the inhibition of IL-6 signaling, may represent a successful second-line agent in ICIs associated with myocarditis, which has important advantages if compared with the other drugs (potential anti-tumorigenic effect and ability to enhance the efficacy of immune checkpoint inhibitors).

Tofacitinib is a first-generation JAK inhibitor, which blocks the production of pro-inflammatory cytokines through the suppression of JAK-pathway (it inhibits JAK3, JAK1, and to a lesser degree, JAK2). It has been developed for the treatment of autoimmune and inflammatory diseases such as rheumatoid arthritis [58]. There is only one case report in the literature regarding the use of oral Tofacitinib in refractory ICI-associated myocarditis. Two patients, after anti-PD-1 antibody therapy for metastatic cancer, developed dyspnea with an elevation of cardiac enzymes, ECG abnormalities, and typical features of myocarditis at cardiac CMR. Both patients responded poorly to the treatment with steroids, plasma exchange, and intravenous immunoglobulin (IVIG), while they had significant and rapid clinical and biochemical improvement with Tofacitinib, at 5 mg twice daily. Tofacitinib may be a new option for the treatment of refractory ICI-associated myocarditis, but it needs more evidence by randomized clinical trials [59].

A recent case report proposes a successful use of statins added to the treatment of ICI-related myocarditis [60]. Three patients with ICI-related myocarditis were treated with intravenous immunoglobulin (IVIG) 1 g/kg/day for 3 days and Rosuvastatin 20 mg daily or Atorvastatin 40 mg daily. They had resolution of cardiological symptoms during a close follow-up and, thus, they resumed anti-cancer therapy. Statins have established immunomodulatory [61] and anti-inflammatory effects [62]. For example, statins exert their immunomodulating effect by decreasing the expression of major histocompatibility complex of class II and therefore decrease the immune response. Interestingly, this is the first case report to describe a patient successfully rechallenged with ICI after ICI-induced myocarditis. This patient had myocarditis with nivolumab and ipilimumab combination therapy, but after treatment with IVIG and statins, he was able to reinitiate with nivolumab. The future challenge will be to obtain a prompt resumption of anticancer therapy (including ICI) for improving the prognosis.

Vitamin D has important immunoregulatory properties, its deficiency and specific genetic polymorphisms of vitamin D receptor (VDR) are associated with major severity of autoimmune diseases. Several studies show that VDR is expressed on immune cells (T-cell, dendritic cell, monocytes, and macrophages) and, thanks to it, vitamin D suppresses T cell proliferation and production of pro-inflammatory mediators [28,63]. A study in mice with induced experimental autoimmune myocarditis (EAM), shows that vitamin D can improve cardiac function, through a reduction of T cell infiltration, myocardial apoptosis, and the number of autophagosomes in cardiac tissue [64]. Furthermore, vitamin D upregulating PD-L1 expression may help to enhance the efficacy of immune checkpoint inhibitors and simultaneously to attenuate IRAEs [65]. Vitamin D has many potential benefits but further investigation is required before it can be used in clinical practice.

## 5. Future Perspectives

Currently, there are little data regarding restarting ICI therapy after myocarditis ICI-related (Figure 2). A recent review [66] and a recent meta-analysis [67] on the safety of ICI re-challenge in patients who underwent treatment cessation due to previous IRAEs suggest that ICIs re-challenge is safe; but the limitations of these studies are that, among IRAEs, they do not include cardiovascular toxicities, such as myocarditis and others toxicities associated with high mortality rate. In fact, in clinical practice, IRAEs such as those involving neurological or cardiovascular systems are not re-challenged.

However, a new observational, cross-sectional, pharmacovigilance cohort study including all IRAEs, such as those associated with high mortality rates, suggests that initial IRAEs considered to be the most life-threatening, including myocarditis and neurological IRAEs, are not associated with higher recurrence rates in rechallenge compared with other initial IRAEs [68].

The American Society of Clinical Oncology recommends holding ICI and permanently discontinuing after grade I of cardiac complications. In the case of I grade (abnormal cardiac biomarker testing, including abnormal ECG) or asymptomatic forms (e.g., an isolated elevation of cardiac troponin without clinically manifest myocarditis), it remains questionable if ICI rechallenge is reasonable for selected cases as in patients with end-stage cancer [35].

The choice to retreat patients after ICI-related cardiovascular toxicity is an important question since it may represent the best and only possible treatment for many advanced cancers.

The decision to carry out a rechallenge, in the absence of an alternative available antineoplastic therapy, depends on the oncological prognosis, the clinical outcome, the severity of the ICI-related myocarditis, the improvement after adequate immunosuppressive therapy, the response to previous therapy, and the patient’s preference in the risk–benefit assessment. ESMO consensus recommendation advises using monotherapy with anti-PD1 if immunotherapy needs to be reintroduced [69].

Therefore, the decision for rechallenge must be considered on clinical judgment on a case-by-case basis with close collaboration between oncologists and cardiologists. Prospective studies are needed to assess the safety of rechallenge and offer potential prophylactic approaches to minimize the risk of severe myocarditis after reintroduction of ICIs.

Another important question is about the primary prevention of myocarditis and the necessity to improve the management of IRAEs without treatment discontinuation. While the interruption of ICIs therapy is necessary to relieve myocarditis and other IRAEs, it can worsen the progression of cancer. In addition to immunotherapy, many other drugs have shown to have a good effect on the management of IRAEs and the research should aim to evaluate the use of them as prophylaxis.

Further studies are needed to better understand the specific pathophysiology and molecular pathways that may have an important role in the pathogenesis of IRAEs. Interesting studies are trying to identify markers capable of predicting the onset of IRAEs such as IL-6, IL-17, G-CSF, to help the diagnosis, management, and surveillance [67].

The use of new cardiac imaging techniques is necessary for an early diagnosis of this potentially fatal complication. Among those, bidimensional echocardiography and newer speckle tracking can help, as it has been demonstrated in various clinical conditions, to diagnose the initial deterioration of longitudinal contractile function of the cardiac chambers, even when ejection fraction, the classical parameter used to evaluate LV systolic function, is not affected.

## Figures and Tables

**Figure 1 biomolecules-11-00785-f001:**
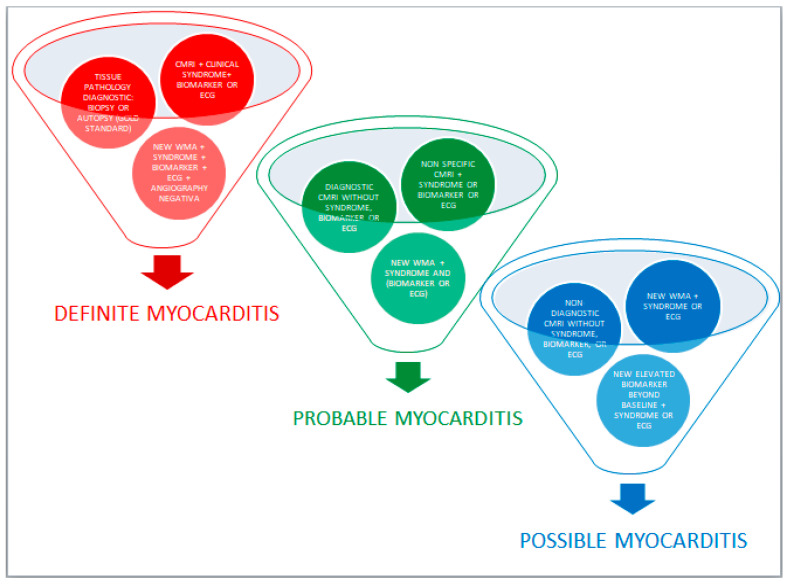
Division of ICI-related myocarditis cases into three groups: definite myocarditis, probable myocarditis, and possible myocarditis.

**Figure 2 biomolecules-11-00785-f002:**
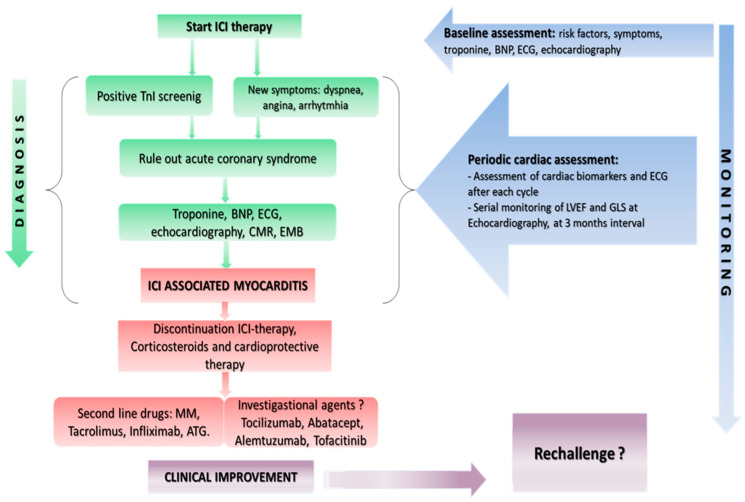
An algorithm for the clinician to make diagnosis of myocarditis and monitoring of cardiovascular toxicity ICI induced. Overview of first-, second-line treatment and investigational agents for ICI-associated cardiotoxicity.

**Table 1 biomolecules-11-00785-t001:** Incidence of cardiotoxicity ICI-associated [13] and proportion of suspected drug-induced cardiovascular events reported with ICIs to the total spectrum of toxicity reactions (IRAEs) [14,15]. VigiBase collects reports of cardiovascular events associated with ICI versus those associated with other drugs. Compared with the full database, ICI treatment was associated with higher reporting of myocarditis (IC_0.25_ 3.20) and pericardial diseases (IC_0.25_ 1.63). [A positive IC_0.25_ value (>0) is the traditional threshold used for statistical significance].

Clinical Trials- Incidence (%) of Cardiovascular IRAEs (22)	VigiBase (WHO Database)-Proportion of Cardiovascular IRAEs to Total IRAEs (18)
Miocarditis 0.09–2.4%	Myocarditis 0.39% (IC_0.25_ 3.2)
Pericarditis < 1–2% Pericardial effusion 2%	Pericardial diseases (pericarditis, pericardial effusion and tamponade) 0.30% (IC_0.25_ 1.63)
Myocardial infarction < 1–2%	Myocardial infarction 0.53% (IC_0.25_ −1.14)
Cardiac Arrhythmia 4%	Supraventricular arrhythmias 0.71% (IC_0.25_ 0.56) Cardiac conductive disorders 0.12% (IC_0.25_ −0.93) Cardiac ventricular arrhythmias 0.07% (IC_0.25_ −2.19)
Heart failure 0.4%	Heart failure 0.72% (IC_0.25_ −0.47)
Takotsubo cardiomyopathy (rarely reported)	Takotsubo cardiomyopathy N/A
Cardiac arrest (rarely reported)	Cardiac death or shock 0.43% (IC_0.25_ −1.28)

**Table 2 biomolecules-11-00785-t002:** Incidence of myocarditis and other cardiotoxic effects for each drug and pharmacological class of ICIs, as reported from clinical trials and within the VigiBase (the World Health Organization global database for ADRs).

Drug	Incidence of Myocarditis	From VigiBase WHO Database [17]				Pharmacological Class	From VigiBase WHO database (from 1 January 2008 to 2 January 2018) [18]			
		Total ADRs	Cardiac ADRs	Proportion of Myocarditis versus all Cardiovascular Events of Each Drug	Pericardial Disease, versus all Cardiovascular Events of Each Drug		Total ADRs	Myocarditis Reported for Each ICIs versus the Full Database	Pericardial Disease Reported for Each ICIs versus the Full Database	Vasculitis Reported for Each ICIs versus the Full Database
Ipilimumab	0.2% [19]	26030	471 (1.81%)	69 (14.6%)	42 (8.92%)	Anti CTLA-4	8266	6 (0.07%)	13 (0.16%)	10 (0.12%)
Nivolumab	0.06% (fatal event <0.01%) [11]	49506	1103 (2.23%)	148 (13.4%)	155 (14.1%)	Anti PD-1 and Anti PD-L1	20643	84 (0.41%)	74 (0.36%)	56 (0.27%)
Pembrolizumab	0.5% [20]	25028	497 (1.99%)	80 (16.1%)	80 (16.1%)					
Atezolizumab	<1% [21]	3627	94 (2.59%)	10 (10.6%)	16 (17%)					
Avelumab	N/A	505	16 (3.17%)	4 (25%)	2 (12.5%)					
Durvalumab	N/A	1329	34 (2.56%)	4 (11.8%)	7 (11.8%)					
Nivolumab and Ipilimumab	0.27% (fatal event 0.17) [11]					Anti PD-1/PD-L1 and anti-CTLA-4	2412	32 (1.33%)	8 (0.33%)	8 (0.33%)

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
