# Peer review of "An Emergent Form of Cardiotoxicity: Acute Myocarditis Induced by Immune Checkpoint Inhibitors"

_biomolecules, 2021, doi:10.3390/biom11060785_

Round 1

Reviewer 1 Report

When it comes to immune check point inhibitors, it would be good to cover the recent progress on PDL1 inhibitors. As currently PDL1 inhibitors are the main class of antibodies that are in clinical trails and are in use also. Also, the authors mostly covered the areas related to myocardia, but ICIs have also been used to treat cancers and also to overcome PDx resistance. Some important immunological areas covering ICIs are missing in this review.

Author Response

Thank you for the great suggestions. We tried to follow them.

Reviewer 2 Report

This is a very interesting and well written review by Esposito et al, which sheds light on the topic of Immune checkpoint inhibitor mediated myocarditis. I would suggest some minor corrections.

  1. On Line 89-94 i would suggest enlarging the discussion on this interesting study. This is especially important for epidemiological insights.
  2. While echocardiography as a baseline investigation is pointed out on Figure 3, it is not mentioned in Chapter 2. It is essential to initially evaluate baseline cardiac function in patients at risk for developing myocarditis due to ICI.

(Also discussed : Oncotarget . 2017 Nov 21;8(63):106165-106166.  doi: 10.18632/oncotarget.22579.  )

  1. On page 6 you discuss Lake Louis criteria from 2009, which were updated in 2018. Please add reference and update lake louis criteria in the text (J Am Coll Cardiol . 2018 Dec 18;72(24):3158-3176. doi: 10.1016/j.jacc.2018.09.072.)
  2. On page 6 Lines 257-258 false negative results of EMB are discussed. Here a reference would be helpful. I would consider discussing explicit indications for EMB. (J Am Coll Cardiol 2018 Dec 18;72(24):3158-3176. doi: 10.1016/j.jacc.2018.09.072.)

Author Response

Thank you for your suggestions, the answers are attached.
